# Polyphasic Validation of a Nisin-Biogel to Control Canine Periodontal Disease

**DOI:** 10.3390/antibiotics9040180

**Published:** 2020-04-14

**Authors:** Eva Cunha, Ferdinando Bernardino Freitas, Berta São Braz, Jorge Moreira da Silva, Luís Tavares, Ana Salomé Veiga, Manuela Oliveira

**Affiliations:** 1CIISA-Centro de Investigação Interdisciplinar em Sanidade Animal, Faculdade de Medicina Veterinária, Universidade de Lisboa, Av. da Universidade Técnica, 1300-477 Lisboa, Portugal; evacunha@fmv.ulisboa.pt (E.C.); ferdinandofreitas@fmv.ulisboa.pt (F.B.F.); bsaobraz@fmv.ulisboa.pt (B.S.B.); ltavares@fmv.ulisboa.pt (L.T.); 2Virbac de Portugal Laboratórios, Lda., Rua do Centro Empresarial, Quinta da Beloura, 2710-693 Sintra, Portugal; jorge.moreira@virbac.pt; 3Instituto de Medicina Molecular, Faculdade de Medicina, Universidade de Lisboa, Avenida Professor Egas Moniz, 1649-028 Lisboa, Portugal; aveiga@medicina.ulisboa.pt

**Keywords:** canine saliva, cell culture, dogs, nisin-biogel, periodontal disease, storage

## Abstract

Background: Periodontal disease (PD) is a highly prevalent inflammatory disease in dogs. This disease is initiated by a polymicrobial biofilm on the teeth surface, whose control includes its prevention and removal. Recently, it was shown that nisin displays antimicrobial activity against canine PD-related bacteria. Moreover, guar gum biogel has shown to be a promising topical delivery system for nisin. Methods: In this study we aimed to evaluate the antimicrobial activity of the nisin-biogel in the presence of canine saliva and after a 24-month storage, at different conditions, using a canine oral enterococci collection. We also studied the nisin-biogel cytotoxicity using a Vero cell line and canine primary intestinal fibroblasts. Results: The presence of saliva hampers nisin-biogel antimicrobial activity, and higher nisin concentrations were required for an effective activity. A significant reduction (*p* ≤ 0.05) in inhibitory activity was observed for nisin-biogel solutions stored at 37 °C, over a 24-month period, which was not observed with the other conditions. The nisin-biogel showed no cytotoxicity against the cells tested at concentrations up to 200 µg/mL. Conclusions: Our results confirmed the potential of the nisin-biogel for canine PD control, supporting the development of an in vivo clinical trial.

## 1. Introduction

Periodontal disease (PD) is a highly prevalent inflammatory disease in dogs, affecting the periodontium tissue [1]. PD is initiated by the formation of a polymicrobial biofilm at the tooth surface, also known as dental plaque, which induces an inflammatory response, leading to gingivitis and/or periodontitis [1]. In addition, bacteria present in the dental plaque may reach the bloodstream and affect distant organs, leading to systemic consequences [2].

Several strategies can be applied to control PD, including home oral hygiene procedures and regular professional periodontal evaluation with the establishment of therapeutic protocols, focusing on the prevention and removal of dental plaque [3]. Recently, an innovative approach based on the antimicrobial peptide nisin, incorporated in the delivery system guar gum gel (nisin-biogel), was demonstrated as a promising alternative for the management of PD in dogs [4]. 

Nisin, a bacteriocin produced mainly by *Lactococcus lactis*, acts through lipid II linkage by physically disrupting the bacterial membrane, with pore formation, and by inhibiting cell wall biosynthesis [4,5,6]. This molecule has antimicrobial activity against Gram-positive bacteria and some Gram-negative bacteria, including several periodontal pathogens [4,5,6,7]. To allow its topical application to the gingival tissue, nisin was incorporated in a guar gum gel, a product that combines the antimicrobial ability of nisin and the delivery ability and antioxidant properties of guar gum, which may be beneficial in PD control [8]. Nevertheless, despite all the advantages of this compound, it should be noted that nisin activity can be influenced by environmental conditions, with several reports referring to the proteolytic degradation of nisin [5,9,10]. To be applied to the oral cavity of dogs, this nisin-biogel should maintain its activity in the presence of canine saliva, which contains enzymes, glycoproteins, immunoglobulins, peptides, inorganic substances, white blood cells, epithelial cells and microorganisms, that may influence nisin’s activity [11]. Additionally, it should not be affected by the pH of saliva, osmolarity or temperature, which may interfere with the native structure of the peptide [9]. 

Another key issue in the development of new therapeutic compounds is the evaluation of their safety regarding eukaryotic cells. Although no information regarding the cytotoxicity of nisin-biogel towards canine cells is available, Shin et al. [7] reported that a 48-h treatment with nisin at 200 µg/mL did not affect the viability of primary human oral cells. 

In this study we developed a polyphasic approach to validate the potential of nisin-biogel for the control of canine PD. First, the influence of canine saliva in the antimicrobial activity of the nisin-biogel towards a previously characterized collection of canine PD enterococci was evaluated, as well as the influence of four storage conditions [12]. Then, the cytotoxicity of nisin-biogel towards two different cell types, a Vero cell line and canine primary small intestinal fibroblasts was determined.

## 2. Results

### 2.1. Antimicrobial Activity of the Nisin-Biogel in the Presence of Dog’s Saliva

Initially we evaluated the antimicrobial activity of nisin-biogel, in the presence of canine saliva, against an oral enterococci collection. Nisin in aqueous solution was used as a control.

In the presence of canine saliva, nisin and nisin-biogel solutions at 12.5 and 25 µg/mL had no antimicrobial activity against any isolate. At 50 µg/mL, nisin solution was able to inhibit one isolate, and the nisin-biogel three isolates. At 100 µg/mL, the nisin and nisin-biogel solutions diluted in saliva presented inhibitory activity against 95% (19/20) and 85% (17/20) of the enterococci, respectively (Table 1). Regarding the controls, saliva and nisin not diluted in saliva at 12.5 µg/mL were not capable of inhibiting any isolate, while the nisin-biogel not diluted in saliva at 25 µg/mL showed antimicrobial activity against six isolates (Table 1). Canine saliva samples presented a pH of 7.7 to 8.

At 100 µg/mL, nisin solutions promoted inhibition halos with diameters ranging between 4.7 and 10 mm, with an average of 7.8 ± 1.3 mm, while the nisin-biogel produced inhibition halos with diameters ranging between 5 and 7.7 mm, with an average of 6.3 ± 0.8 mm. 

A statistical difference (*p*-value = 0.0001) was observed between the inhibition zone diameters produced by nisin and nisin-biogel solutions.

### 2.2. Viability of the Nisin-Biogel under Different Storage Conditions

The antimicrobial activity of nisin-biogel, submitted to four storage temperatures (−20 °C, 0–4 °C, room temperature and 37 °C) over a 24-month period was evaluated. Nisin in aqueous solution was used as a control.

Regarding the evaluation of the effect of storage, the mean inhibition zone diameters obtained for nisin and nisin-biogel solutions, at different storage conditions, over a 24-month period are presented in Figure 1. Statistical analysis showed no significant differences (*p*-value > 0.05) in the diameters of the inhibition halos promoted by the solutions of nisin or nisin-biogel. Regarding the effect of storage temperature over the 24-month period, it was possible to observe that the differences between the diameter of the inhibition halos produced by the solutions stored at 37 °C and by the solutions stored at other temperatures were statistically different (*p*-value ≤ 0.05).

### 2.3. Cytotoxicity Activity of the Nisin-Biogel

The effect of nisin and nisin-biogel on cell viability was evaluated using two cell types, a Vero cell line, used as a control, and canine primary small intestinal fibroblast cells, to represent the canine gastrointestinal environment [13]. Nisin solutions were used as controls.

Exposition to nisin solutions ranging from 5 to 400 µg/mL was slightly or non-cytotoxic for the Vero cell line, independently of the exposure time (Figure 2a), except for the nisin solution at 400 µg/mL after 24 h that was severely cytotoxic. Additionally, the nisin solution at 600 µg/mL was moderately cytotoxic after 30 min of exposure, and severely cytotoxic after 12 h and 24 h of exposure (Figure 2a). Comparing to the control cells, with no nisin treatment, significant statistical differences (*p*-value ≤ 0.05) were observed in all exposure times regarding the nisin solution of 400 and 600 µg/mL. 

To the contrary, the nisin-biogel showed non or slight cytotoxicity in all doses and exposure times tested (Figure 2b). Compared to the control cells, a significant statistical difference (*p*-value ≤ 0.05) in viability was observed at 30 min with nisin-biogel at 400 µg/mL.

Regarding the canine primary cells, nisin solutions ranging from 5 to 400 µg/mL were non-cytotoxic after 30 min of exposure, while the nisin solution at 600 µg/mL was moderately cytotoxic, with significant statistical difference (*p*-value ≤ 0.05) when compared to the non-treated cells (Figure 3a). After 12 and 24 h of exposure, nisin solutions ranging from 5 to 200 µg/mL were slightly or non-cytotoxic, while nisin solutions at 400 and 600 µg/mL were moderately cytotoxic, with significant statistical difference (*p*-value ≤ 0.05) (Figure 3a). Moreover, nisin-biogel solutions showed non-toxicity in all doses and exposure times tested (Figure 3b).

Considering the evaluation of the exposure time of Vero cells, it was possible to observe that for nisin solutions, the values obtained with 30 min of exposure were statistically different from 12 and 24 h (*p* < 0.0001); and for nisin-biogel solutions all exposures times were statistically different (*p* < 0.0001). Regarding canine primary cells, results after exposure to nisin and nisin-biogel solutions were not statistically different in the three exposure times.

Additionally, evaluating the effect of biogel in the cells viability, it was possible to observe that in the Vero cell line no significant statistical difference (*p*-value > 0.05) was observed between solutions, while for canine primary cells, absorbance values obtained for nisin solutions and nisin-biogel solutions were statistically different (*p*-value < 0.0001).

## 3. Discussion

Periodontal disease is one of the most prevalent inflammatory diseases in dogs, being the prevention of dental plaque formation an important step for its control [1]. Recently, our laboratory proposed a promising strategy for PD control involving the use of the bacteriocin nisin incorporated in a guar gum biogel as a delivery system (nisin-biogel) [4,6,7,14]. However, the influence of the oral environment in nisin activity could be a cornerstone for a potential clinical application. The influence of canine saliva on nisin and nisin-biogel antimicrobial activity regarding canine PD enterococci was evaluated, being observed that the presence of saliva increased the required concentration for nisin’s inhibitory activity, when compared with the minimum inhibitory concentrations (MIC) of nisin and nisin-biogel previously determined [4]. More precisely, for the nisin solution, a concentration eight-fold higher than the MIC was required to promote an inhibitory activity against 95% (19/20) of the isolates, while for nisin-biogel, a concentration four-fold higher than the MIC was needed to inhibit 85% (17/20) of the tested strains. Despite Tong et al. [15], Shin et al. [7] have stated that nisin’s activity is not influenced by human saliva, and there are several reported differences between canine and human saliva, the pH being one of the major differences [5,7,11,15,16]. Humans have a salivary pH ranging from 6.2 to 7.4, while dogs present a more alkaline pH, ranging between 7 to 9 [17,18]. Nisin is a small cationic antimicrobial peptide whose structural stability and antimicrobial activity are influenced by environmental conditions, like pH, being more stable and effective at acidic conditions [9]. More specifically, when the pH is higher than its isoelectric point (pH > 8) several irreversible structural modifications in the nisin molecule occur [9]. Therefore, the pH values of the canine saliva samples used in this study may have limited nisin’s activity. In addition, it is important to refer to the proteomic composition of canine and human saliva, which also diverges, including differences in several components such as enzymes, glycoproteins, immunoglobulins or peptides [11,19]. Additionally, nisin can be inactivated by proteolytic activity, but degradation by salivary enzymes has not been described [5,6,9,15].

Regarding nisin-biogel solutions, the inhibition zone diameters obtained with biogel solutions were more consistent when compared with nisin solutions, revealing a potential effect of the biogel on stabilizing nisin diffusion, further confirming the potential of the guar gum gel as a nisin topical delivery system.

A viability assay was also performed, allowing us to identify the optimal temperature and storage period to guarantee the compound antimicrobial efficacy. Our results revealed that nisin and nisin-biogel solutions kept their antimicrobial activity in all temperatures and timepoints evaluated, except for the nisin solution of 500 µg/mL at 37 °C in the last timepoint (Figure 1). Besides that, the storage temperature of 37 °C caused a statistically significant decrease in nisin activity in all tested solutions, compared with the remaining temperatures, over the 24-month storage period (Figure 1). These results suggest that 37 °C storage is not recommended as a storage condition for nisin or nisin-biogel solutions. In fact, reports state that nisin’s activity is kept at a high range of temperature, but its thermostability is reduced with an increasing pH. Moreover, nisin is highly stable at freezing temperatures and after autoclaving at 121 °C in a pH 2 solution, but it is completely inactivated after 30 min at 63 °C in a pH 11 solution [9]. In addition to this, no statistical differences were observed between inhibition halos produced by nisin or nisin-biogel solution. In fact, guar gum gel is a natural polysaccharide, stable at a wide range of temperatures, however, high temperatures can disturb guar gum conformation, reducing its viscosity, which may have occurred at 37 °C storage [20]. 

Finally, to evaluate the effect of the nisin-biogel on cell viability, two cell types were used. The cytotoxicity evaluation of potential oral topical products is indispensable because of their close contact with the gingiva, enamel, cementoenamel junction and the oral mucosa. Cationic antimicrobial peptides can cause different effects on different eukaryotic cells, and some reports have already evaluated nisin toxicity effects on different eukaryotic cells [7,21,22,23]. In this study, a Vero cell line was used as the control and primary canine small intestinal fibroblast cells were used to understand the potential toxicity of the solutions regarding canine intestinal cells. Although nisin can suffer proteolytic degradation by gastrointestinal enzymes, such as pancreatin and α-chymotrypsin [9], its incorporation in the guar gum biogel can protect it from being completely released in the oral cavity, allowing nisin to progress to the gastrointestinal tract. Then in the gastrointestinal environment, a proteolytic degradation of the bacteriocin is expected, but biogel may have a protective effect on nisin [24]. For this reason, the cytotoxicity assay was also performed on canine small intestinal cells.

Our study showed an absence of toxicity (cell viability higher than 90%) with nisin and nisin-biogel solutions, in both cell types, until 200 µg/mL.

Regarding the Vero cell line, nisin solutions until 200 µg/mL were non cytotoxic, but at 400 µg/mL a slightly cytotoxicity was observed after 30 min and 12h, and severe toxicity after 24h of exposure. Additionally, nisin solution at 600 µg/mL showed a moderate toxicity after 30 min and severe toxicity after 12 h and 24 h of exposure. Different results were observed by Vaucher and collaborators in 2010, which evaluated the cytotoxicity of nisin on Vero cells after 24 h of exposure, obtaining a 50% reduction on viability with nisin concentration of 0.35 µg/mL [23]. Additionally, nisin-biogel solutions were non-toxic to Vero cells until 200 µg/mL, being slightly toxic at 400 µg/mL in all exposure periods.

Canine primary cells showed more than 90% viability, after all exposure times, to all nisin-biogel solutions (Figure 3b), while nisin solution at 600 µg/mL, in all exposure times, and at 400 µg/mL, after 12 h and 24 h of exposure, presented moderate cytotoxicity. Results obtained were similar to previous studies performed by Shin et al. [7], which showed that nisin at 100 µg/mL was non-cytotoxic to primary periodontal ligament cells, gingival fibroblast cells, primary human oral keratinocytes and osteoblast- like cells, with cells revealing a normal attachment and proliferation capacity [7]. Only at nisin concentrations superior to 200 µg/mL and after 48 h of exposure, cells started to show low levels of apoptosis [7]. Additionally, similar to our study, in 2006 Maher and McClean evaluated nisin toxicity regarding two types of human intestinal cells, following 24 h of exposure, and reported a 50% reduction in viability with concentrations ranging from 89.9 to 115 µM, which correspond to 301 to 386 µg/mL [22].

Together, results show that the nisin-biogel seems to be an appropriate approach for PD control in dogs. In the future its antimicrobial activity against other periodontal bacteria from the canine dental plaque and against polymicrobial biofilms should be evaluated. Finally, in vivo studies evaluating the efficacy and safety of nisin-biogel would be essential for developing a commercial product for clinical application.

## 4. Materials and Methods

### 4.1. Bacterial Strains 

A collection of 20 oral enterococci, including 17 *Enterococcus faecalis* isolates and 3 *Enterococcus faecium* isolates, previous characterized regarding clonality, antimicrobial resistance and virulence profiles, were used as bacterial models [4,12]. All enterococci were obtained from the oral cavity of dogs with PD and included planktonic and biofilm-producer strains [12]. *Enterococcus faecalis* ATCC^®^ 29212 was used as a control strain.

### 4.2. Nisin Preparation

A nisin stock solution (1000 µg/mL) was prepared according to Santos et al. [25] by dissolving 1 g of nisin powder (2.5% purity, 1000 IU/mg, Sigma-Aldrich, St. Louis, MO, USA) in 25 mL of HCl (0.02 M) (Merck, Darmstadt, Germany), followed by filtration using a 0.22 µm Millipore filter (Frilabo, Maia, Portugal) [25]. After, serial dilutions were prepared using distilled sterile water, which were kept at 4 °C during the study.

### 4.3. Biogel Preparation

A 1.5% guar gum gel (*w/v*) solution was obtained by dissolving 0.75 g of guar gum (Sigma-Aldrich, St. Louis, MO, USA) in 50 mL of distilled sterile water. Then, the solution was sterilized by autoclave, and nisin was incorporated within the guar gum gel (biogel) in a proportion of 1:1, to obtain a 0.75% biogel (*w/v*) [25].

### 4.4. Collection of Dog’s Saliva

Saliva was collected in a Portuguese veterinary hospital, from healthy dogs after the owner’s consent. Samples were collected using a sterile Pasteur pipette and placed into sterilized containers. Afterwards, the collected saliva was filtered using a 0.22 µm Millipore filter (Frilabo, Maia, Portugal) and kept at −20 °C [15]. Saliva pH was measured using a pH indicator paper.

### 4.5. Antimicrobial Activity of the Nisin-Biogel in the Presence of Dog’s Saliva

To evaluate the influence of canine saliva in the antimicrobial activity of the nisin-biogel, a spot-on-lawn assay was performed [15]. To optimize salivary enzymatic activity, saliva was incubated for one hour at 37 °C [15]. Then, nisin-biogel was diluted in saliva, in a proportion of 1:1, to obtain the following nisin concentrations: 100, 50, 25, and 12.5 µg/mL [4]. Additionally, nisin diluted in saliva (100, 50, 25, and 12.5 µg/mL), nisin diluted in sterile distilled water (12.5 µg/mL), nisin-biogel (25 µg/mL) and saliva were included as controls [4]. A 10^7^ CFU/mL bacterial suspension was prepared for each isolate, evenly spread onto the surface of Brain Heart Infusion (BHI) agar plates (VWR, Leuven, Belgium) using a sterile swab, after which 10 µL of every solution of nisin and nisin-biogel diluted in saliva or sterile distilled water were spotted onto the same agar plates. After a 24 h incubation at 37 °C, all plates were observed for the detection of inhibition halos, which diameters were measured. To ensure the biological relevance of the results all experiments were performed in triplicate on independent days.

### 4.6. Viability of the Nisin-Biogel under Different Storage Conditions

The spot-on-lawn assay was also used to evaluate the effect of storage conditions in the antimicrobial activity of nisin-biogel. Nisin solutions were included as controls. Three solutions of nisin and nisin-biogel at different concentrations (62.5, 250 and 500 µg/mL) were stored at different temperatures (−20 °C, 0–4 °C, room temperature and 37 °C) for 24 months. The antimicrobial activity of the stored solutions was tested at nine different time storage points (1, 3, 6, 9, 12, 15, 18, 21 and 24 months), using the same four bacterial strains (M2b, M4c, M29b and 29212) randomly selected from our collection of canine PD enterococci [4]. For this evaluation a 10^8^ CFU/mL bacterial suspension was prepared for each isolate and spread onto the surface of BHI agar plates. Then, a 3 μL drop of each nisin and nisin-biogel solutions under testing were spotted on the plates, followed by incubation at 37 °C for 24 h, and measurement of inhibition zone diameters.

### 4.7. Cytotoxicity Activity of the nisin-Biogel

The cytotoxicity potential of the nisin-biogel against eukaryotic cells was accessed using a control Vero cell line and a canine primary small intestinal fibroblast cell culture (d-6025 Cell Biologics^®^, Chicago, IL, USA). Nisin solutions were included as controls. Cells were maintained in Dulbecco’s Modified Eagle Medium (DMEM, LONZA, Basel, Switzerland ) supplemented with 10% Fetal Bovine Serum (Biowest, Riverside, MO, USA), 1% penicillin/streptomycin (VWR, Leuven, Belgium) and 1% fungizone (VWR, Leuven, Belgium), at 37 °C in a humidified 5% CO_2_ atmosphere. Fresh medium was provided every 3 days, until cells reached 80% confluency, observed by an inverted microscope (Olympus CK30). Cell passages two to five were used in the cytotoxicity assay. Cells were counted using a Neubauer chamber after dilution in Trypan Blue Dye (VWR, Leuven, Belgium), after which 1 × 10^4^ cells per well were plated in a 96-well microplate (Thermo Scientific, Waltham, MA, USA) [7]. After 24 h of incubation at 37 °C in a humidified 5% CO_2_ atmosphere, cells were exposed to 10 to 30 µL of nisin (final concentrations: 5, 12.5, 25, 50, 75, 100, 200, 400 and 600 µg/mL) and 20 to 60 µL of nisin-biogel (final concentrations: 12.5, 25, 50, 75, 100, 200 and 400 µg/mL) diluted in fresh medium, and incubated through three exposure times: 30 min, 12 h and 24 h [4,7]. Negative controls, with only medium and non-supplemented biogel (gg), were also included. Following each assay, cells viability was evaluated by a colorimetric assay using tetrazolium salt thiazolyl blue 3-(4,5-dimethylthiazol-2-yl)-2,5-diphenyltetrazolium bromide (VWR, Leuven, Belgium), also known as MTT assay [13,26,27]. Briefly, after the exposure time, 100 µL of a 10% MTT solution (5 mg/mL) dissolved in fresh medium was added to each well and the plates were incubated for 3 h. The MTT was then removed and 100 μL of dimethyl sulfoxide (PanReac AppliChem, Barcelona, Spain) was added to each well to dissolve the formazan crystals. Then, dose- and time-dependent cell death was evaluated by determination of the optical density (OD) at 570 nm and cell viability was calculated according to the formula (Equation (1)) [13,22,28,29,30]: (1)$$\mathrm{Cell}\;\mathrm{viability}\;\left(\%\right)=\frac{OD570\;of\;the\;sample}{OD570\;of\;the\;control}\times 100$$

According to the cell viability, the tested solutions were classified as non-cytotoxic (more than 90% cell viability), slightly cytotoxic (60–90% cell viability), moderately cytotoxic (30–59% cell viability) and severely cytotoxic (less than 30% cell viability) [13,28,29]. These experiments were performed in triplicate on independent days.

### 4.8. Statistical Analysis

Statistical analysis was carried out using GraphPad Prism 5^®^ (Graphpad Software, San Diego, CA, USA) and Microsoft Excel 2016^®^. Student’s t test was used for statistical analysis between the inhibitory zone diameters promoted by nisin and nisin-biogel solutions diluted in saliva. Storage effect on antimicrobial activity of nisin and nisin-biogel was evaluated by Student’s t test and linear regression. Cells viability were analyzed by Student’s t test and ANOVA, with the post hoc test Tukey’s Multiple Comparison Test.

Quantitative variables were expressed as mean values ± standard deviation. A confidence interval of 95% was considered, with a *p*-value ≤ 0.05 indicating statistical significance.

## Figures and Tables

**Figure 1 antibiotics-09-00180-f001:**
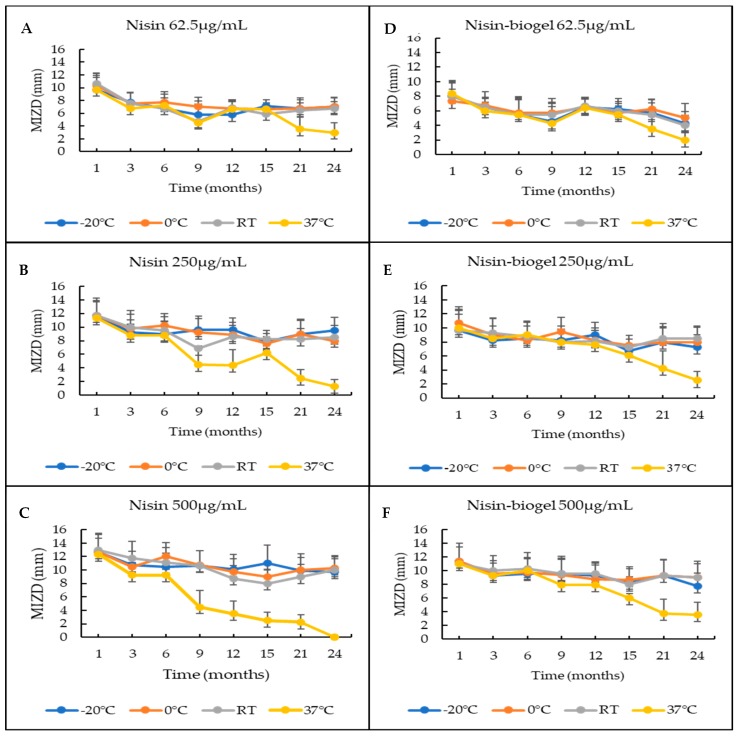
Mean inhibition zone diameters (MIZD) promoted by nisin and nisin-biogel solutions, over a 24 month period at four storage conditions (–20 °C, 0–4 °C, RT-room temperature and 37 °C). The vertical bars represent the standard error of the mean. (**A**)-MIZD promoted by nisin at 62.5 µg/mL; (**B**)-MIZD promoted by nisin at 250 µg/mL; (**C**)-MIZD promoted by nisin at 500 µg/mL; (**D**)-MIZD promoted by nisin-biogel at 62.5 µg/mL; (**E**)-MIZD promoted by nisin-biogel at 250 µg/mL; (**F**)-MIZD promoted by nisin-biogel at 500 µg/mL

**Figure 2 antibiotics-09-00180-f002:**
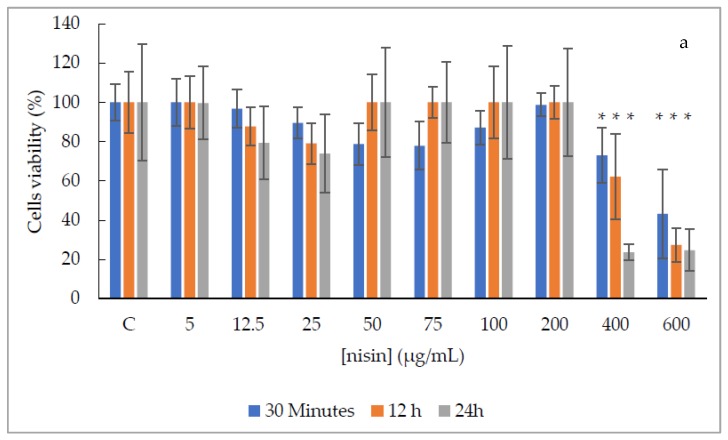
Vero cell line viability (%), after three treatment periods (30 min, 12 h and 24 h) with nisin and nisin-biogel solutions. (**a**)—Distribution of cells viability (%) after exposure to nisin at concentrations ranging between 5 to 600 µg/mL, after three treatment periods. (**b**)—Distribution of cells viability (%) after exposure to nisin-biogel at concentrations ranging between 12.5 to 400 µg/mL, after three treatment periods. C—negative control with no treatment. gg—control with non-supplemented biogel. The vertical bars represent standard deviations of means * *p*-value ≤ 0.05.

**Figure 3 antibiotics-09-00180-f003:**
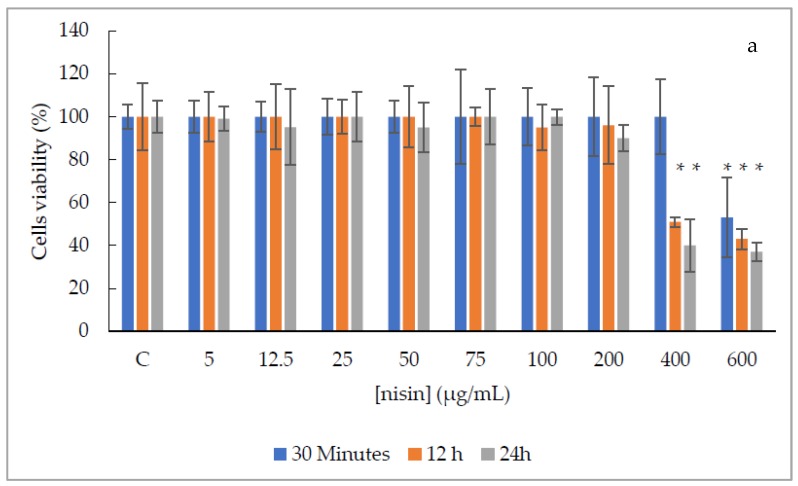
Canine primary small intestinal fibroblast cells viability (%), after three treatment periods (30 min, 12 h and 24 h) with nisin and nisin-biogel solutions. (**a**)—Distribution of cells viability (%) after exposure to nisin at concentrations ranging between 5 to 600 µg/mL, after three treatment periods. (**b**)—Distribution of cells viability (%) after exposure to nisin-biogel at concentrations ranging between 12.5 to 400 µg/mL after three treatment periods. C—negative control with no treatment. gg—control with non-supplemented biogel. The vertical bars represent standard deviations of the means * *p*-value ≤ 0.05.

**Table 1 antibiotics-09-00180-t001:** Distribution of the number of isolates inhibited by the nisin and nisin-biogel solutions, diluted or not in saliva. Saliva was also included as a control.

Formulation		[nisin] µg/mL	Number of Isolates Inhibited
	Saliva	-	0
In the absence of saliva	nisin	12.5	0
	nisin-biogel	25	6
In the presence of saliva	nisin	12.5	0
		25	0
		50	1
		100	19
	nisin-biogel	12.5	0
		25	0
		50	3
		100	17

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
