# Peer review of "Polyphasic Validation of a Nisin-Biogel to Control Canine Periodontal Disease"

_antibiotics, 2020, doi:10.3390/antibiotics9040180_

Round 1

Reviewer 1 Report

This contribution to a Special Issue by Cunha et al. builds incrementally on previous work from the same group that highlights the potential of a nisin-guar gum gel formulation for treatment of periodontal disease in dogs. There are three novel sets of experiments: (i) It is shown that teatment with canine saliva has little effect on the efficacy of nicin as an antibacterial peptide against clinical isolates of Enterococcus spp. (ii) It is shown that the nisin gel is as stable as nisin itself to storage over 24 months at room temperature (or lower temperatures), showing only little loss of efficacy. (iii) It is shown that that the nisin gel has little or no cytoxicity towards mammalian cells in culture at concentrations of nisin that would be used therapeutically. These are data that will probably contribute to an interesting clinical trial in the future.

I have just a few minor comments that should be addressed in revision.

  1. Although I have recommended some improvement in English expression and grammar, the manuscript is already quite readable.
  2. Line 222: "viability with nisin concentration of 50 of 0.35 μg/mL [23]": I do not understand what is meant here. Please clarify. Also, do the authors have an explanation for the large difference in cytotoxicity between their data and those in reference [23]?
  3. Lines 272 and 287: Please indicate the total cfu spread to obtain the lawns (or the volumes used).
  4. Line 286: "using four bacterial strains randomly selected from our collection of canine PD enterococci". Were the same four strains used throughout for all experiments. or were they randomly selected for each experiment/time point? Please clarify. Furthermore, since some strains were insensitive to nisin/nisin gel at the lowest concentration used (Table 1), these must have been excluded, meaning that the selection was not truly "random". Maybe it would be best to actually list the four strains used (they are characterized in the previous paper, ref. [4]). If there are data for four separate strains, then error bars should be shown in Figure 1 and explained in the legend.
  5. Line 327: The data in Figure 1 were analysed by linear regression - is there a reason to believe that nisin should lose activity in a linear (rather than exponential) manner with time?

Author Response

We would like to thank you for the comments and the discussion points presented. We agree that these points will improve the quality and understanding of the manuscript. So, we analyse every comment and suggestion and we have proceeded to its clarification, as you can see in the manuscript and in this letter.
As requested and to facilitate the reading, reviewer reports and respective answers are listed below.

Although I have recommended some improvement in English expression and grammar, the manuscript is already quite readable.

We have revised the grammar throughout the document.

  1. Line 222: "viability with nisin concentration of 50 of 0.35 μg/mL [23]": I do not understand what is meant here. Please clarify. Also, do the authors have an explanation for the large difference in cytotoxicity between their data and those in reference [23]?

We understand your remark, as there was an error in line 222. The correct sentence should be “viability with nisin concentration of 0.35 μg/mL”. In fact, results presented by reference [23] are quite different from ours, while most studies that evaluated the cytotoxicity of nisin have obtained results similar to the ones from our study. We believe that the differences in the methodology used by authors of reference [23] may be responsible for the differences observed. For example, in our study we only introduced nisin and nisin-biogel formulations in the 96-well microplate 24 hours after cells incubation, unlike reference [23]. Also, in reference [23] there is no information regarding the volume of nisin used in the cytotoxic assay. Also, nisin stock solution was prepared differently. All these points may justify the observed differences.

  1. Lines 272 and 287: Please indicate the total cfu spread to obtain the lawns (or the volumes used).

As mentioned in line 271, a 107 CFU/mL bacterial suspension was prepared for each isolate. Afterwards, these suspensions were evenly spread onto the surface of Brain Heart Infusion agar plates with a swab. We included this information in the manuscript.

  1. Line 286: "using four bacterial strains randomly selected from our collection of canine PD enterococci". Were the same four strains used throughout for all experiments. or were they randomly selected for each experiment/time point? Please clarify. Furthermore, since some strains were insensitive to nisin/nisin gel at the lowest concentration used (Table 1), these must have been excluded, meaning that the selection was not truly "random". Maybe it would be best to actually list the four strains used (they are characterized in the previous paper, ref. [4]). If there are data for four separate strains, then error bars should be shown in Figure 1 and explained in the legend.

We understand your remark. Initially, four bacterial strains were randomly selected from our collection, and used throughout the 24-month assay. As suggested, we have included strains identification in the text (lines 285/286). Also, we have changed Figure 1 as suggested, and a new Figure 1 was added to the manuscript, including changes suggested by both reviewers.

  1. Line 327: The data in Figure 1 were analysed by linear regression - is there a reason to believe that nisin should lose activity in a linear (rather than exponential) manner with time?

According to our knowledge, as the storage period increased we expected to observe a gradual decrease in nisin’s activity. In our study, the decrease in nisin’s activity resulted in a reduction of the mean inhibition zone diameters, although results from some time points analysed were not in line with this tendency. During data evaluation, it was possible to observe that a linear function was the more adequate method to compare results from each storage condition.

Reviewer 2 Report

The manuscript by Eva Cunha and coworkers gives a few pointers on the antimicrobial activity of the peptide nisin, incorporated in the delivery system guar gum gel (nisin-biogel) in the presence of canine saliva and after a 24-month storage under three different storage temperatures. Nisin-biogel cytotoxicity was also evaluated on a Vero cell line (again, a study against this cell line was already reported) and canine primary intestinal fibroblasts.

The results reported in this work are modest and of small significance, but they might be of interest to readers of Antibiotics working on nisin-related research projects. Therefore I would recommend publication after some, minor revisions, as listed below.

Abstract. The following sentence seems to contradict itself: "The presence of saliva did not inhibit the nisin-biogel antimicrobial activity, but higher nisin concentrations were required for an effective activity." Please, rephrase it. Maybe "inhibit" can be replaced by "hamper"? By the way, please correct "were".

Table 1. The concentration at which nisin-biogel is able to inhibit three or more isolates without saliva should be added for comparison. The value should be available from literature (e.g., ref. 4). If so, the related reference should be added.

p.2, two lines above Table 1. "the nisin-biogel not diluted in saliva at 25 μg/mL showed antimicrobial activity against six isolates". This is in contrast with the value reported in the Table. Please, correct the values in the table (or in the text).

p.2, last line before Table 1. "Canine saliva samples presented a pH between 7.7 and 8". How did the authors estimate such a close interval by means of a simple pH indicator paper (as stated in the Experimental session)?

p.3, last paragraph. Please, add the pH of the stored samples, in view of your comments reported in the Discussion section (p.8, first paragraph) on the key role played by pH on nisin stability.

Figure 1 is misleading. Please, use exactly the same scale on y axis (with the same maximum) on each and every graph (it cannot be 12 vs. 10, or 14 vs. 12, etc.).

p.5, first line. Please, justify the choice of using Vero cell line, being it not dog-related.

Figures 2a and 2b. Vero fibroblast cells?

p.5 five lines form the top: "However, the nisin solution at 600 μg/mL was moderately cytotoxic after 30 minutes of exposure, and severely cytotoxic after 12 and 24 hours of exposure (Figure 2a)". The cytotoxic effect doesn't seem so dramatically different over time at 600 μg/mL. Conversely, the comment fits 400 μg/mL much better.

Discussion, line 177. "several irreversible structural modifications". Are those  chemical modifications rather than 3D-structural modification?

Conclusions. I would remove this paragraph altogether because it doesn't add anything to the paper and it mostly refers to literature. The Discussion section is enough.

Author Response

We would like to thank you for the comments and the discussion points presented. We agree that these points will improve the quality and understanding of the manuscript. So, we analyse every comment and suggestion and we have proceeded to its clarification, as you can see in the manuscript and in this letter.
As requested and to facilitate the reading, reviewer reports and respective answers are listed below.

  1. The following sentence seems to contradict itself: "The presence of saliva did not inhibit thenisin-biogel antimicrobial activity, but higher nisin concentrations were required for an effective activity." Please, rephrase it. Maybe "inhibit" can be replaced by "hamper"? By the way, please correct "were".

According to your suggestion we replaced “inhibit” by “hamper” and corrected the word “were” (lines 23/24).

  1. Table 1. The concentration at which nisin-biogel is able to inhibit three or more isolates without saliva should be added for comparison. The value should be available from literature (e.g., ref. 4). If so, the related reference should be added.

As mentioned in lines 74-76, “saliva and nisin not diluted in saliva at 12.5 µg/mL were not capable of inhibit any isolate, while the nisin-biogel not diluted in saliva at 25 µg/mL showed antimicrobial activity against six isolates”. This information is presented in the corrected Table 1.

The work presented in reference [4] has shown that all isolates used in this study were susceptible to the nisin-biogel, and the concentrations used were selected based on the reference [4], as described in lines 270-272.

  1. 2, two lines above Table 1. "the nisin-biogel not diluted in saliva at 25 μg/mL showed antimicrobial activity against six isolates". This is in contrast with the value reported in the Table. Please, correct the values in the table (or in the text).

We thank you for your observation, as there were an error in Table 1. The number of isolates inhibited by nisin at 12.5 µg/mL is 0, and the number of isolates inhibited by the nisin-biogel at 25 µg/mL is 6. Table 1 was corrected accordingly.

  1. 2, last line before Table 1. "Canine saliva samples presented a pH between 7.7 and 8". How did the authors estimate such a close interval by means of a simple pH indicator paper (as stated in the Experimental session)?

The determination of the saliva samples pH was based on the colour variation of the indicator paper used, by comparison with the colour reference list. According to the scale of the pH indicator paper used, solutions with pH values between 7.7 and 8 present a specific colour, easily recognized by direct observation.

  1. 3, last paragraph. Please, add the pH of the stored samples, in view of your comments reported in the Discussion section (p.8, first paragraph) on the key role played by pH on nisin stability.

We thank you for your comment. In fact, pH is described as an important factor that can influence nisin activity, as described in the discussion. We have determined the initial pH of the solutions used in this study (which was found to be between 2 and 3), but we did not determined the pH values of all the solutions tested over time.

  1. Figure 1 is misleading. Please, use exactly the same scale on y axis (with the same maximum) on each and every graph (it cannot be 12 vs. 10, or 14 vs. 12, etc.).

Figure 1 was changed according to the suggestions of both reviewers.

  1. 5, first line. Please, justify the choice of using Vero cell line, being it not dog-related.

The Vero cell line was used as a control to evaluate the cytotoxicity of our formulations. This cell line has well-defined culturing characteristics in experimental settings and its usually used for cytotoxicity evaluation of several drugs, including of molecules used in odontology. In addition, there was no information available regarding nisin-biogel cytotoxicity towards this cell line, and therefore we believed to be important to perform this evaluation.

  1. Figures 2a and 2b. Vero fibroblast cells?

The legend of Figure 2a and 2b was changed according to suggestion, with “Vero fibroblast cells” being replaced by “Vero cell line” (lines 123 and 127).

  1. 5 five lines form the top: "However, the nisin solution at 600 μg/mL was moderately cytotoxic after 30 minutes of exposure, and severely cytotoxic after 12 and 24 hours of exposure (Figure 2a)". The cytotoxic effect doesn't seem so dramatically different over time at 600 μg/mL. Conversely, the comment fits 400 μg/mL much better.

We agree with your comment and included information regarding cytotoxicity results of nisin solution at 400 µg/mL in lines 112/113.

  1. Discussion, line 177. "several irreversible structural modifications". Are those chemical modifications rather than 3D-structural modification?

We are referring to chemical changes as previously described by [9].

Conclusions. I would remove this paragraph altogether because it doesn't add anything to the paper and it mostly refers to literature. The Discussion section is enough.

According to suggestion, we have removed the conclusion paragraph of the manuscript.
